# Can We Talk about Smart Tourist Villages in Mărginimea Sibiului, Romania?

**George Moise** ¹, **Agatha Popescu** ², **Iulian Alexandru Bratu** ¹, **Ion Răducuță** ³,*, **Bogdan Gabriel Nistoreanu** ⁴ **and Mirela Stanciu** ¹,*

¹ Faculty of Agricultural Sciences, Food Industry and Environmental Protection, "Lucian Blaga" University of Sibiu, 5-7 Ion Ratiu Street, 550003 Sibiu, Romania; george.moise@ulbsibiu.ro (G.M.); iulian.bratu@ulbsibiu.ro (I.A.B.)

² Faculty of Management and Rural Development, University of Agronomic Sciences and Veterinary Medicine Bucharest, 59 Mărăști Boulevard, District 1, 011464 Bucharest, Romania

³ Faculty of Animal Productions Engineering and Management, University of Agronomic Sciences and Veterinary Medicine Bucharest, 59 Mărăști Boulevard, District 1, 011464 Bucharest, Romania

⁴ Faculty of Business Administration in Foreign Languages, Academy of Economic Studies, Piața Romană, 010374 Bucharest, Romania

* Correspondence: ion.raducuta@usamv.ro (I.R.); mirela.stanciu@ulbsibiu.ro (M.S.)

**Abstract:** The aim of this paper is to evaluate some villages in the mountain area of Sibiu County, Romania, as "smart" tourist villages. The research was carried out in two stages: Stage 1 was collection of information on (a) food products registered in different national and international quality schemes and (b) the number of tourist structures by type, comfort category, and accommodation capacity. Stage 2 was a case study conducted among 32 rural tourism entrepreneurs from 3 villages in Mărginimea Sibiului, using a questionnaire with 22 items on the criteria for evaluation of "smart" villages. The answers were systematized, statistically processed, and interpreted. The main results are as follows: (a) Concerning products, producers, and gastronomic points (PGL) registered, there are 9 products in the quality scheme "mountain product", 10 producers on the platform "Gusturisibiene", 1 producer of "telemea de Sibiu", a registered product with protected geographical indication, and 2 PGL. (b) The number of officially registered tourist structures is 93, of which 72% are agritourism guesthouses. (c) The results for the evaluation of villages as "smart" are that 78.2% of locals use the internet, 74% own at least one smart device, 60.16% of energy used is renewable. There are low values of water and air pollution; there are local job and business opportunities (93.75%); future local development activities will target agriculture with all its sectors and agritourism; public administrations have implemented 7 types of activities to increase the attractiveness of localities; and quality of life, services, and social life aspects are perceived to be at high standards.

**Keywords:** smart village; smart tourism; smart communities; rural development; Mărginimea Sibiului; Romania





## 1. Introduction

At the end of 2021, the population of the European Union was estimated at 446.8 million inhabitants. Depopulation of rural areas is one of the problems facing Europe. A rural population statistical data set for 47 European countries between 1960 and 2021 shows that the rural population weight varies widely, from 85.53% in Liechtenstein to 1.88% in Belgium and 0% in Gibraltar and Monaco. Romania, with a 45.67% rural population, ranks 6th in this hierarchy.

The European Smart Villages Forum states that the term smart village (SVs) refers to people and communities that are more active, self-reliant, and resilient. Thus, starting from the five pillars of agriculture and food, energy and green mobility, economy, social inclusion, and digitization, solutions are proposed to transform villages for a sustainable future [1].

At the EU level, the countryside covers more than 80% of the territory of the European Union ([2], pp. 1–3) and provides a living environment for about 137 million inhabitants, who represent around 30% of the total population. The major challenges facing Europe's rural areas are depopulation; an ageing population; a changing rural landscape; and the need for multifunctionality of farms [3]. All these force us to find more inclusive and sustainable future solutions [4].

According to data provided by the World Tourism Organization, in 2022, Europe recorded about 508 million arrivals, or about 80% of the 2019 level [5].

A recently published report [6] shows that with a Digital Economy and Society Index value of 30.6 points compared to the EU average of 52.3 points in 2022, Romania ranks 27th out of 27 EU Member States. This index covers four areas: digital human capital skills, connectivity, digital technology integration, and digital public services. In terms of human capital and its digital skills, Romania faces a lack of basic digital skills among its population. Romania scores well below the EU average in at least basic digital skills (28% vs. 54%) and digital skills above basic level (9% vs. 26%). The share of use of fixed broadband coverage of at least 100 Mbps (57%) and of very high-capacity fixed networks is 87% in Romania, which is above the EU average. Romania's connectivity score is 55.2 (15th out of 27 countries), compared to 59.9 on average in the EU. For digital technology integration, Romania's score is 15.2 (27th out of 27 countries), compared to an EU average score of 36.1. Furthermore, with a score of 21 for digital public services, Romania ranks 27th in the EU (EU average score 67.3).

The smart village (SV) concept involves the digital transformation of key activities in rural areas with the aim of revitalizing and strengthening local communities [7]. The transformation and modernization of rural tourism is made possible by the emergence of new innovative technologies.

The main purpose of this paper is to evaluate some villages in the mountainous area of Sibiu County, Romania, as "smart" tourist villages. "Smart" tourist villages are well-established communities that highlight local strengths and opportunities and use innovative solutions and state-of-the-art technologies, with the ultimate goal of developing local tourism and improving the quality of life of their inhabitants.

The research is structured in the following sections: Section 1 is the introduction, Section 2 conveys the literature review, Section 3 explains the methodology, Section 4 introduces the case study, Section 5 comprises the discussions, and Section 6 includes the main conclusions.

## 2. Literature Review

### 2.1. Smart Village

Smart villages (SVs) are made up of rural communities who seek innovative and practical solutions to the challenges they face and who adapt quickly to opportunities for rural transformation ([2], pp. 9–12). The construction of smart villages must be developed in harmony with environmental protection and the sociocultural and economic needs of the community, and it must be integrated into the regional development strategy [8]. SVs aim for an inclusive development, following all the objectives of the European Green Pact [9].

Advances in digital technology have influenced many aspects of our daily life and, implicitly, the emergence and implementation of the concept of SVs [10–12]. SVs use innovative solutions to increase the quality of life [13]. Innovative technologies that can be used by smart village communities are the Internet of Things, Artificial Intelligence, Big Data, blockchain, use of nanomaterials, drones, and robots. They can be used to improve many aspects of the life and work of rural residents [14]. Examples of services in rural areas using digital technologies are intelligent transport systems, resource optimization, better waste management solutions, environmental quality control, energy and lighting solutions, precision agriculture, etc. The implementation of new digital technologies and better connectivity in rural areas would also allow the implementation of new and modern forms of learning such as e-learning and virtual courses.

One of the Green Deal objectives is that by 2025, all households in rural areas should have 100% access to fast broadband internet [15]. In 2019, only 59% of rural households had access to broadband internet (compared to 87% of EU households). The share of digital literacy among the EU rural population was 48% in 2019 [16].

The SV concept is multidisciplinary in nature and has grown as an area of interest in international research in recent years [17]. The smart villages model includes, but is not limited to, the following areas: health, education, economy, environment, sustainable development, digital transformation, renewable energy, healthy food, awareness, and civic engagement [18]. Aziiza and Susanto [19] propose a smart village model structured on six tiers: governance, technology, resources, services, living and tourism. The aspects included in the SV model are shown in Figure 1.

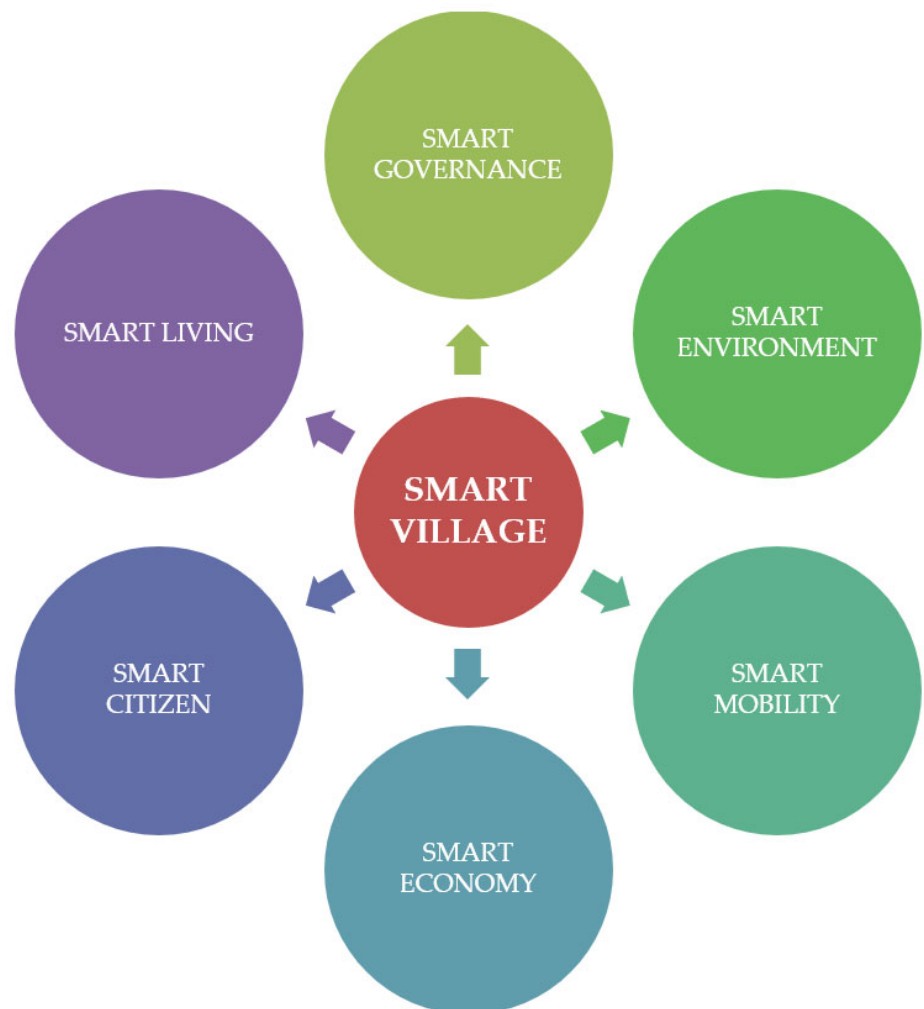

**Figure 1.** The SV Concept, own design, after [19].

In China, Li and Zhong [20] conducted a study to propose a model for the development of historic tourist villages based on the smart village concept. The authors included the following groups of factors in the model: technological; economic; culture and tourism; environmental; and social. They concluded that these factor groups are interlinked, most of them requiring a sustainable development direction.

Wang Q. et al. [21] propose a dynamic development model of the concept of SVs, at the core of which is the knowledge-based rural society, managed by the local community.

Following the implementation between 2018 and 2021 of a project entitled "Smart village" by organizations in seven countries, some recommendations were made to the policy environment, including prioritizing digital transformation of rural areas; inclusion

of the concept of SV in current and future strategies and policies; and development of the necessary digital infrastructure and associated skills [22].

### 2.2. Smart Tourist Villages

Ciolac et al. [23] perceive the smart tourist village from the perspective of sustainable entrepreneurship, while Jeftic et al. [24] believe that small rural communities should be considered as development hubs. Such communities can increase the rural population retention and stimulate healthy food production, gastronomy being a key element in the development of a tourist destination [25].

The economic, social, and environmental benefits of agritourism motivate the local community to implement and develop it, considering that in recent decades, the demand for and supply of agritourism have increased [26]. In general, landscapes, agricultural activities, and agricultural food products have a positive influence in the creation of touristic products and services [27]. In 2022, Obeidat B. [28] showed that improving the demand for local products, promoting cultural exchange, experience opportunities, and improving the image of the region are vital for the success of agritourism. In general, agritourism requires the involvement of tourists in various activities closely related to the work of farmers [29] who practice extensive agriculture, contributing to the long-term synergy between rural tourism and agriculture [30] and the sustainability of the community [31]. Sustainable tourism development contributes to the preservation of the agricultural cultural heritage.

SVs use energy from alternative sources and apply waste treatment technologies to maintain a healthy living environment.

The practice of ecotourism [32] and agritourism [33,34] in communities bordering protected areas can be a model of sustainable use [34] of the tourism resources that the area offers [35,36]. Increased investment in building a smart rural tourism environment leads to improved living conditions for locals and tourists and facilitates access to good-quality public services [37], while providing ecosystem services [38]. The satisfaction of tourists depends on the technologies used, the existing innovations, and the quality of the local experience [39].

### 2.3. Smart Tourism

The tourism sector is among the areas that have rapidly adopted the Internet of Things (IoT) to create innovative services. The results of a survey of Spanish tourists by Ballina [40] show that they appreciate the existence of technology innovations in rural tourism destinations. Intelligent models for developing rural tourism must be based on digital technologies [41].

Various papers present smart apps [42] and gamification [43] as means of education, interactive discovery, and engagement in the many activities that generate a memorable holiday experience in smart tourism villages [44]. Suanpang et al. [45] show that the metaverse enables the creation of virtual tourism experiences in smart destinations as an integral part of sustainable tourism.

Tourism (with its multiple forms of manifestation) in rural areas in general and in disadvantaged areas in particular is sustainable if it meets the following conditions: it is diffuse; it respects the natural and cultural rural heritage; it involves the active participation of the local population; it maintains the traditional activities of the area; it commercializes authentic and quality tourism products; and it directly markets tourism products.

The advantages and risks that the development of various forms of tourism brings to disadvantaged rural areas are presented in Table 1.

**Table 1.** Benefits and associated risks of tourism development (in its many forms) in disadvantaged areas (own design).

| Benefits of Different Forms of Tourism for Areas in Decline | Risks that Tourism can Bring to Areas in Decline |
|---|---|
| **Economic level** | |
| Job creation and income generation complementary to agriculture | Congestion of the area |
| Diversification of the local economy | Danger of monoactivity |
| Maintain and improve local services and activities | |
| Improving the professional skills of local people and entrepreneurs | |
| Direct valorization of local products and creation of short food chains | |
| Improving the general infrastructure of the locality | Creating additional infrastructure and/or services may lead to loss of authenticity |
| **Environmental level** | |
| Beautifying the locality | Increased risk of air, water, soil pollution Increasing the risk of noise pollution |
| Maintenance, protection, and improvement of natural areas | Risk of disturbance to local flora and fauna due to excessive visitor numbers |
| **Anthropological level** | |
| Preserving traditional farming and animal breeding practices | |
| Gastronomy based on local, fresh, seasonal products and local recipes | Loss of local identity and depersonalization |
| Preservation of local crafts | Destruction of local architecture |
| Maintains and enhances local customs and traditions | Altering and changing local customs and traditions |
| Increasing local community interest in recreational and cultural activities | |
| Improving the life quality of residents | |
| Encouraging local entrepreneurship | |
| Allows sociocultural exchanges with tourists | Risk of conflict between the local community and tourists Too many tourists can lead to anti-social behavior |

## 3. Data and Methodology

Based on the premise that there are major challenges for rural communities and rural public authorities in boosting development and improving living conditions for residents [46], we chose to analyze how rural tourism and agritourism entrepreneurs in three communities in Mărginimea Sibiului, Romania, perceive their own villages as smart tourist villages.

### 3.1. Research Objectives and Steps of the Study

We used the case study method that was elaborated in the first stage using desk research methods and employing national statistics and databases of lodging, food, and travel services provided by the National Authority for Tourism under the coordination of the ministry in charge of tourism activities or by the Ministry of Agriculture and Rural Development.

In the second stage of the research, a sociological survey was carried out and a questionnaire was developed as a working tool. The questionnaire was pretested with 4 guesthouse owners (1 from Sibiel, 1 from Gura Râului, and 2 from Rășinari). The final questionnaire included 22 items related to the potential of localities to be or become smart villages and 4 items on the socio-demographic data of the respondents. Face-to-face interviews were

carried out between 1 February and 7 March 2023 with 32 owners/managers of tourist structures with accommodation functions (12 from Sibiel, 10 from Gura Râului, and 10 from Rășinari) in 3 localities in Mărginimea Sibiului, chosen because of the large number of tourist structures. The duration of each interview was around 25 min/interviewee. Respondents were assured of the confidentiality of their answers and their use for scientific purposes.

The answers were synthesized and statistically processed using Excel, v. 365, Microsoft Corporation, Redmond, WA, USA.

The frequencies and the shares of the answers were processed using the following:

- *Semantic Differential Scale* (Osgood, C.E., 1957), which reflects the intensity of the opinions based on the weighted arithmetic mean for each item of a questionnaire according to the formula:

$$\sum_{i=1}^{n} x_i\, f_i\, /\, \sum_{i=1}^{n} f_i \tag{1}$$

where $x_i$ is the score connected to the appreciation, and $f_i$ is the frequency, or, more exactly, the number of answers registered for each score.

- *Likert Scale* (1932), which reflects the agreement and disagreement of the respondents related to an item of a questionnaire. In the article were used both 4 and 5 Point Likert scale. The results were illustrated in suggestive graphics and tables and have been correspondingly interpreted. Finally, the main conclusions were drawn.

The objectives and the research steps are presented in Table 2.

**Table 2.** Methodological scheme of the research.

| Research Steps | Research Objectives |
|---|---|
| Theoretical research | • Obtaining information on the number of agricultural products registered in different national quality schemes |
| | • Obtain information on the number of tourist facilities by type, level of comfort, and accommodation capacity |
| Case study on "smart tourist village" | • Identify the perception of the acceptance of the village as a smart tourist village |
| | • Analysis of the transport infrastructure in the three localities |
| | • Identification of local development issues and satisfaction regarding quality of life and services |
| | • Identification of innovative technology solutions used in tourist facilities and the origin of food destined for tourists |
| | • Identify the perception of the general appearance of the locality and the efforts made to preserve the natural and anthropic tourist heritage |
| Discussion and main conclusions | • Outlining the main future directions for smart tourism development |

### 3.2. Presentation of the Studied Area

The three tourist villages studied are part of the Mărginimea Sibiului area, emblematic for pastoral civilization. In this area, the landscape of a traditional mountain village and the authenticity of the culinary experiences are harmoniously combined. In 2015, the whole area was awarded the title of "European Destination of Excellence for Tourism and Gastronomy" by the European Commission. The Sibiu region also received the title of

"European Gastronomic Region" in 2019, because of the interaction between cultural and ethnic diversity, traditional agriculture, and the desire for innovation of the locals [47].

A project entitled "Revitalizing Remote and Mountainous areas through Sustainable Alternative Tourism" is currently underway to promote the "Mărginimea Sibiului" tourism area. It is funded through the Interreg Europe program, which has the objective of implementing ecotourism, adventure, and cultural and rural tourism in this area [48].

The specific folk traditions, local and international festivals, Romanian costumes, hospitality, local architecture, the picturesque landscape, and the specific cuisine recognized thanks to the traditional agricultural products and the old tradition of cheese production make Mărginimea Sibiului a special area. The locations within the Mărginimea Sibiului area of the three tourist villages chosen as case studies are shown in Figure 2.

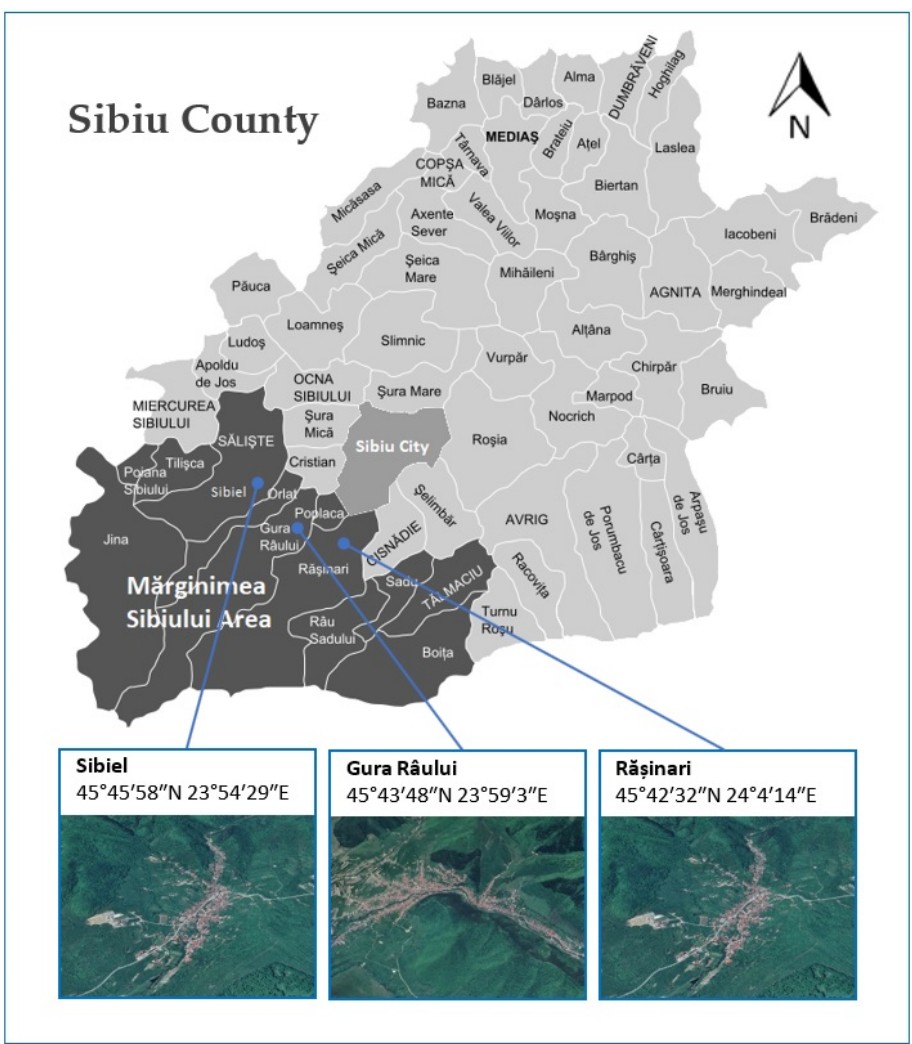

**Figure 2.** Mărginimea Sibiului and locations of the three tourist villages chosen for the study. Source: own design, according to [48].

**The village of Sibiel** is located 20 km from Sibiu, is a typical mountain village, and has been nominated as a tourist village since 1972. In 1964, the priest Zosim Oancea became the parish priest of the village and started a collection of icons painted on glass, which was the basis of the current museum of icons painted on glass by popular craftsmen. The collection of icons initiated by the village priest has become a point of attraction for tens of thousands of tourists from home and abroad. Between 1970 and 1990, the village was visited by tourists from Israel and the USA, who dined or stayed with the farmers, confirming its pioneering status in the field of agritourism. Sibiel hosts an annual fair

to promote local varieties of apple and pear trees from orchards over 100 years old. The beauty of the mountain landscape, the well-preserved traditions, the local gastronomy, and traditional farming practices, together with the Roman vestiges in the surrounding area, known as the "Salgo fortress", have made the village of Sibiel a reference point for agritourism in the Mărginimea Sibiului since 1990.

**Gura Râului** is a compact village at an altitude of 554 m, located in the mountain area near the Cindrel Mountains, 18 km from Sibiu. It has been documented since 1380 and has a population of 3621 inhabitants. At the entrance to the commune of Gura-Râului, you are greeted by the beauty of the mountains and the whole environment that seems to embrace the locality. In the past, there were numerous artisanal plants in the locality that used the force of water to extract cold oil or process wool or wood. Some of these are still in operation and can be visited in peasant households. In order to promote local traditions, since 2004, the local public administration has been organizing the Peacock Festival at the beginning of July. The event takes place over two days, starting with a traditional shepherd's dinner, followed on the next day by a game in the village center and a folklore show. In August, a local celebration called "Îmbracă-te românește" is organized to ensure the continuity of the traditional wear.

**Rășinari Commune**, 12 km away from the county town of Sibiu, is the village in Mărginimea Sibiului with the oldest documentary evidence in the county, dating back to 1204. The locality is in the mountain area, having been famous in the past for its pastoral character, and currently has a population of 5416 inhabitants [49]. The promotion of the pastoral tradition and the tourist potential of the locality are carried out through the organization of numerous events, such as the Cheese and Brandy Festival, the Cabbage Man Day, a marathon entitled "Urme pe plai" (Traces on the mountain paths), and the cultural event "Open the gates to the legends of the Rășinari". The stories and legends of the village can be discovered by tourists with the help of a mobile technology application "Questo", which offers a "treasure hunt" through which sights and tourist routes can be identified [50]. The local public administration in partnership with the Sibiu County Council and the County Tourism Association have marked numerous tourist routes in the surroundings of the locality. The action is part of the "Years of Hiking" Program [51]. Through a project called "Let's go with us on the fortress", the local public administration has managed to set up a tourist stop and mark trails to the hill of the same name [52]. As a recognition of the efforts made to promote the tourist heritage, Rășinari Commune received at the end of 2022 from the World Tourism Organization the title of "Best tourism village" [53], being the first village in Romania included in this ranking.

The specific local activities in these villages are ruminant farming, wood processing, and traditional meat processing. The Sibiu County Tourism Association has created a trail in Mărginimea Sibiului called the "cheese trail" with the aim of linking the area's pastoral tradition with tourism [54].

Sibiu was the European Capital of Culture in 2007, and both the city and several rural communities were promoted through numerous events. On this occasion, the Sibiu Regional Ecomuseum Association was founded, which is made up of members from six rural communities, including Gura Râului and Săliște, with the neighboring village of Sibiel.

## 4. Results

### 4.1. Number of Local Products Registered in Various Local, National, and European Quality Schemes and Tourist Accommodation Facilities by Category, Comfort Level, and Capacity

4.1.1. Number of Agri-Food Products Registered in Various National and European Quality Schemes

The Ministry of Agriculture and Rural Development promotes food products registered in various national and European quality schemes. In the catalogue of certified products and activities at the national level, in 2023, there were 3876 entries, of which 158 were from producers in Sibiu County [55]. It is to be noted that the producers of Sibiu "telemea" cheese (cottage cheese) have associated themselves and have managed to obtain

recognition of their product on the European quality scheme with a Protected Geographical Indication, and among the producers of the association, there is one who is domiciled in the municipality of Rășinari.

The Mountain Area Agency of Romania has created the "mountain product" platform on which producers from the mountain area are registered. Its aim is to transmit important messages to buyers about the quality of food and the added value of food products produced in the mountain area. On the national platform "mountain product", for Sibiu County, 148 products are registered, but only 9 of them (8 fruit categories and 1 honey assortment) [56] belong to producers in the 3 localities chosen for the study.

Sibiu County Council supports local producers and promotes a healthy lifestyle. Its sustainable development department has created a platform called "taste of Sibiu", which presents 213 local producers, grouped in 6 micro-regions. Maps have been created showing their addresses by locality [57]. On this platform, we have identified nine producers in the locality of Rășinari selling vegetables, fruit, eggs, dairy, and apricot products and one producer in the village of Sibiel offering eggs and dairy products for sale.

In order to support farmers in the mountain area, since 2018, local gastronomic points have been authorized in these localities, with a maximum capacity of 12 people. In these gastronomic establishments, the raw material used comes mainly from their own production and the recipes are specific to the area and the season. In Sibiu County, 21 local gastronomic points have been established from 2018–2022 [58], of which only two are operating in Gura Râului.

4.1.2. Number of Tourist Accommodation Structures in the Three Localities, Accommodation Capacity, and Degree of Comfort

In the 3 rural localities of Mărginimea Sibiului chosen as a case study, there are currently 93 tourist structures with accommodation functions, as shown in Tables 3–5.

**Table 3.** Number of tourist structures, degree of comfort, and accommodation capacity in Rășinari.

| Type of Touristic Structure | Number | Degree of Comfort (Stars/Flowers) | | | | | Number of Rooms | Accommodation Places |
|---|---|---|---|---|---|---|---|---|
| | | 1 | 2 | 3 | 4 | 5 | | |
| Touristic guesthouse | 20 | | 4 | 13 | 1 | 2 | 159 | 292 |
| Agritouristic guesthouse | 1 | | 1 | | | | 2 | 4 |
| Cottage | 1 | | | 1 | | | 19 | 36 |
| Villa | 3 | | | 2 | 1 | | 50 | 99 |
| Hostel | 1 | 1 | | | | | 5 | 7 |
| Rooms for rent | 3 | | 2 | 1 | | | 17 | 34 |
| TOTAL | 29 | 1 | 7 | 17 | 2 | 2 | 252 | 472 |

Source: own calculation according to [59], consulted on 15 February 2023.

**Table 4.** Number of tourist structures, degree of comfort, and accommodation capacity in Sibiel.

| Type of Touristic Structure | Number | Degree of Comfort (Stars/Flowers) | | | | | Number of Rooms | Accommodation Places |
|---|---|---|---|---|---|---|---|---|
| | | 1 | 2 | 3 | 4 | 5 | | |
| Touristic guesthouse | 26 | | 12 | 10 | 2 | 2 | 135 | 290 |
| Agritouristic guesthouse | 1 | | | 1 | | | 8 | 16 |
| Hotel | 1 | | | 1 | | | 21 | 42 |
| Camping huts | 1 | | | | 1 | | 8 | 16 |
| Rooms for rent | 2 | | 2 | | | | 7 | 14 |
| TOTAL | 31 | | 14 | 12 | 3 | 2 | 179 | 378 |

Source: own calculation according to [59], consulted on 15 February 2023.

**Table 5.** Number of tourist structures, degree of comfort, and accommodation capacity in Gura Râului.

| Type of Touristic Structure | Number | Degree of Comfort (Stars/Flowers) | | | | | Number of Rooms | Accommodation Places |
|---|---|---|---|---|---|---|---|---|
| | | 1 | 2 | 3 | 4 | 5 | | |
| Touristic guesthouse | 21 | | 7 | 11 | 3 | | 134 | 289 |
| Agritouristic guesthouse | 2 | | | 2 | | | 11 | 26 |
| Hotel | 1 | | | | 1 | | 30 | 61 |
| Cottage | 5 | | | 5 | | | 21 | 46 |
| Rooms for rent | 4 | 1 | | 3 | | | 27 | 58 |
| TOTAL | 33 | 1 | 7 | 21 | 4 | | 223 | 480 |

Source: own calculation according to [59], consulted on 15 February 2023.

From the total of 93 tourist accommodation structures registered in the three localities, 67 (72%) are rural tourist hostels. The total accommodation capacity of the 93 tourist structures is 654 rooms and 1330 places. It should be noted that 50 of them (53.80%) are in the three-star/daisies comfort category and only around 14% in the higher-comfort categories. According to the data provided by the National Institute of Statistics, the net utilization index of the tourist accommodation capacity in operation in Sibiu County in August 2022 was 31.7% in the case of tourist guesthouses and 30.5% in the case of agritourism guesthouses, compared to 19.9% and 17.1%, respectively, nationally [60]. The numbers of tourist arrivals in 2022 in the three localities were 5789 (Rășinari), 7732 (Gura Răului), and 14,306 (Sibiel) [61], and the numbers of overnight stays were 11,414 (Rășinari), 14,627 (Gura Răului), and 23,487 (Sibiel) [62].

Leisure activities offered to tourists include archery, hiking, fishing, campfires, jacuzzies, saunas, children's playgrounds, local gastronomy, children's camps, wagon rides, and events.

*4.2. Case Study Conducted among the Owners of Tourist Accommodation Structures in Three Villages in Mărginimea Sibiului*

4.2.1. Socio-Demographic Characteristics of the Group

The main characteristics of the owners of tourist accommodations in the three villages are presented in Table 6.

**Table 6.** Characteristics of the owners of tourist structures in the study area (n = 32).

| Variable | Specification | No | % | Variable | Specification | No | % |
|---|---|---|---|---|---|---|---|
| Gender | Female | 20 | 62.5 | | High school | 15 | 46.9 |
| | Male | 12 | 37.5 | Education | Post-secondary school | 3 | 9.4 |
| | 21–25 | 1 | 3.1 | | Faculty | 11 | 34.3 |
| | 41–45 | 6 | 18.8 | | Master's | 3 | 9.4 |
| Age (years) | 46–50 | 3 | 94 | Employment status | Owner | 19 | 59.3 |
| | 51–55 | 11 | 34.4 | | Employed in state structures and owner | 11 | 34.3 |
| | 56–60 | 5 | 15.6 | | Retirees | 2 | 6.3 |
| | Over 60 | 6 | 18.7 | | | | |

Most of the owners of tourist structures in the 3 localities surveyed are women (62.50%), 22 (68.75%) of them are over 51 years old, 14 people (43.75%) have higher education at university and/or master's level, 59.30% consider themselves entrepreneurs in rural tourism, and 40.7% of them also earn income from salaries or pensions.

4.2.2. Identifying the Perception of Owners of Tourist Structures Regarding the Acceptance of Their Locality as a "Smart Tourist Village"

In order to achieve the aim of the research, 32 owners of tourist accommodation structures in Sibiel, Gura Râului, and Rășinari were interviewed. The owners of tourist guesthouses identified several types of services available in the locality (Table 7).

**Table 7.** Availability of basic services, accessibility for residents, and their perception as being modern and innovative.

| Types of Services at Locality Level | Availability of Services | | Accessibility of Services | | Modern and Innovative Services | |
|---|---|---|---|---|---|---|
| | **Yes** | **No** | **Yes** | **No** | **Yes** | **No** |
| Primary school | 32 | - | 32 | - | 32 | - |
| % | 100 | - | 100 | - | 100 | |
| Secondary school | 20 | 12 | 19 | 13 | 19 | 13 |
| % | 62.5 | 37.5 | 59.4 | 40.6 | 59.4 | 40.6 |
| High school | 1 | 31 | - | 32 | - | 32 |
| % | 3.2 | 96.8 | - | 100 | - | 100 |
| Human medical dispensary | 18 | 14 | 20 | 12 | 20 | 12 |
| % | 56 | 44 | 62.5 | 37.5 | 62.5 | 37.5 |
| Veterinary medical dispensary | 10 | 22 | 10 | 22 | 10 | 22 |
| % | 31 | 69 | 31.3 | 68.7 | 31.3 | 68.7 |
| Hospital | - | 32 | - | 32 | - | 32 |
| % | - | 100 | - | 100 | - | 100 |
| Renewable energy sources | 32 | | 32 | - | 32 | - |
| % | 100 | - | 100 | - | 100 | - |
| Drinking water supply | 32 | - | 32 | - | 32 | - |
| % | 100 | - | 100 | | 100 | - |
| Waste management and sorting | 32 | - | 32 | - | 32 | - |
| % | 100 | - | 100 | | 100 | - |
| Public transport | 32 | - | 32 | - | 32 | - |
| % | 100 | - | 100 | | 100 | - |
| Alternative transport: bikes, scooters, electric cars | 30 | 2 | 30 | 2 | 30 | 2 |
| % | 93.8 | 6.2 | 93.8 | 6.2 | 93.8 | 6.2 |
| Infrastructure for walking and cycling | 30 | 2 | 30 | 2 | 30 | 2 |
| % | 93.8 | 6.2 | 93.8 | 6.2 | 93.8 | 6.2 |

Source: own calculation based on field research.

It is noted that 37.5% said that there is no secondary school in the locality, 96.8% that there is no high school, 44% said that there is no human medical dispensary, and 69% said that there is no veterinary medical dispensary. The perception of the accessibility of services in the locality is closely related to the identification of the existence of these services. The main modern and innovative services are perceived to be education at primary (100%) and secondary school levels (59.4%), medical services provided by the human dispensary (62.5%), the existence of renewable energy sources (100%), drinking water supply (100%), waste management and sorting (100%), public transport (100%), alternative transport, and infrastructure for walking and cycling.

The surveyed hostel managers believe that 100% of them have access to fast internet, modern digital equipment (smart phone, laptop, or tablet), and possibilities to connect to online e-commerce, telemedicine, or e-learning services in their localities. According to their perception (Table 8), the share of the local population using the internet is 78.2% (ranging from 74.6% in Sibiel to 85.7% in Gura Râului), 74% own at least one electronic device that allows them to connect to the internet (ranging from 60.8% in Sibiel to 89.7% in Gura Râului), 46.9% access online banking services (varying from 22.5% in Sibiel to 69% in Gura Râului), and 7.7% follow online professional training courses (varying from 4.75% in Sibiel to 12.3% in Rășinari).

**Table 8.** The respondents' perception of the share of the local population using the internet and digital technologies.

| Localities | Share of Population That: | | | |
|---|---|---|---|---|
| | Use the Internet (%) | Own at Least Smartphone, Laptop, or Tablet | Access Online Banking | Follow Online Training Courses |
| Sibiel | 74.60 | 60.80 | 22.50 | 4.75 |
| Gura Râului | 85.70 | 89.70 | 69 | 6.60 |
| Rășinari | 75 | 74 | 54 | 12.3 |
| Average | 78.20 | 74 | 46.90 | 7.7 |

Source: own calculation based on field research.

To improve living conditions, authorities and residents are turning to innovative technologies. The main sources of heating energy that the administrators of tourist structures know are used in their locality are conventional energy (100%), solar panels (100%), photovoltaic panels (58.4% in Sibiel and 100% in the other localities), charcoal (60% in Gura Râului and 0% in the other localities), wood in 100% proportion, and sawdust pellets (50% in Sibiel and 100% in the other localities). The 32 administrators of tourist structures believe that 60.16% of their localities use renewable energy sources, ranging from 51.5% in Gura Râului to 74.5% in Rășinari. At the same time, 53% of respondents believe that smart farming technologies (precision farming, drones, smart tractors, etc.) are used in their localities.

When asked if they have heard of any environmental protection measures/programs being implemented or being considered in their locality, 90.6% know of such measures to improve energy efficiency, 100% have heard of measures to reduce carbon emissions, 25% have heard of the implementation of water management systems, and 43.75% have heard of a desire to improve transport accessibility.

The degree of air pollution in the locality is perceived as very low by 81.25% of respondents and low by 15.63% of them. In the case of water pollution, 56.25% of respondents perceive the pollution as very low, and 37.5% perceive it as low (Figure 3).

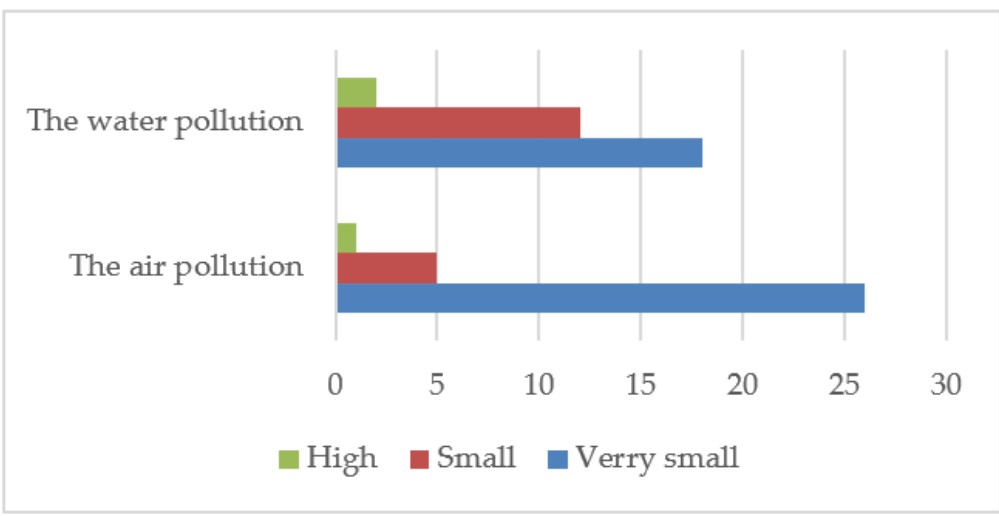

**Figure 3.** Perception of local air and water pollution. Source: own calculation based on field research.

4.2.3. Analysis of the Transport Infrastructure in the Three Localities

The average travel time by car from the village to the county town is between 10 min (Rășinari) and 30 min (Gura Râului). Apart from personal cars, to travel from the village to the center of the county town, 68.75% of the inhabitants use minibuses for commuters, public transport buses are used by 100% in Gura Râului and Rășinari and 0% in Sibiel,

trains are used by 100% in Sibiel, and trams are used by 100% in Rășinari for touristic purpose.

According to the respondents' perception, the main measures that local public authorities should take to improve transport accessibility differ according to the locality (Figure 4).

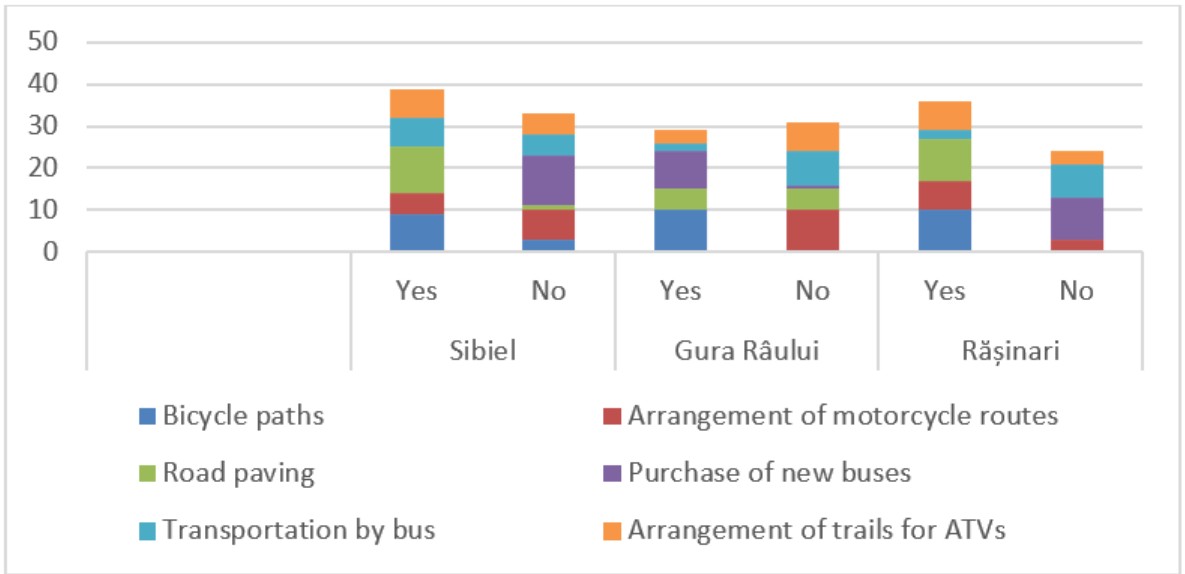

**Figure 4.** Measures to improve mobility and accessibility at local level. Source: own calculation based on field research.

In Sibiel, 91.7% of respondents think that roads need to be paved and 75% highlight the need for bike lanes. In Gura Râului, all respondents think that bicycle lanes should be installed, and 90% highlight the need to purchase new buses. In Rășinari, all respondents say that bicycle lanes and asphalting of roads are needed.

Overall, 90.6% of the respondents would like to see bicycle paths, 81.25% would like to see asphalted roads, 53% would like to see ATV trails, and 37.5% would like to see motorcycle trails.

4.2.4. Identification of Local Economic Development Issues and Satisfaction Regarding Quality of Life and Services

- Although almost all respondents believe that job opportunities (93.75%) and local businesses (100%) exist in their locality, 34.37% fail to identify programs or initiatives to support the local economy.
- Respondents differently identified important economic sectors in the three localities. In Sibiel, agritourism is the most important activity, followed by animal husbandry, fruit growing, agriculture, and vegetable growing, as the locality has an agricultural profile. In Gura Râului, agritourism, animal breeding, fish farming, meat processing, and wood processing are very important, as well as milk processing and agriculture. Local producers have set up small workshops for the traditional processing of pork and sheep meat, and some of them have their own facilities for the primary processing of wood into building materials. In Rășinari, 10 types of activities are nominated as being of equal importance: agriculture, apiculture, animal breeding, the existence of on-farm slaughtering points, milk and meat processing, wood processing, trade of general products, and small industry workshops. In addition to the tradition of sheep breeding, the locals are also renowned for their commercial spirit. The respondents from this locality did not identify agritourism as one of the important economic sectors in the locality.

Respondents indicated the share of inhabitants who have a job, the share of existing businesses at local level, and the average income per inhabitant (Figure 5).

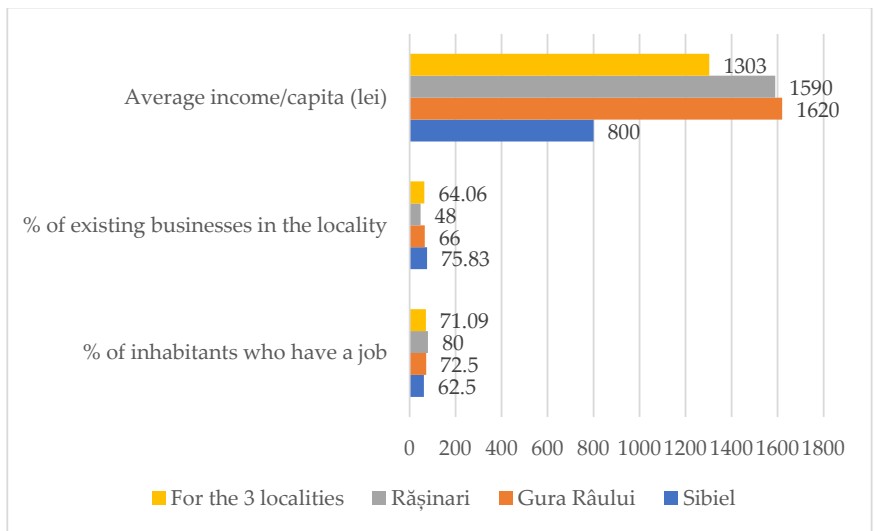

**Figure 5.** Respondents' perceptions of local development. (Exchange rate on 20 March 2023 according to the National Bank of Romania: 1 euro = 4.9219 lei.) Source: own design based on field research.

According to the opinion of the respondents in the three tourist villages, 71.09% of the locals have a job (with a minimum value of 62.5% in Sibiel and a maximum of 80% in Rășinari), around 64.06% of small local businesses are registered (with a minimum value of 48% in Rășinari and a maximum of 75.83% in Sibiel), and the average income per inhabitant is around 1303 lei (with a minimum value of 800 lei in Sibiel and a maximum of 1620 lei in Gura Râului). These results correlate with the image of Sibiel as an agricultural village and Gura Râului as a village with small local businesses.

Respondents believe that local public administrations have implemented seven types of activities aimed at increasing the attractiveness of the locality, stimulating the development of the local economy, and creating jobs and businesses, with differences depending on the general specifics of the village (Figure 6).

The three tourist villages are promoted in the media, and there are festivals, local product fairs, and local cultural events. Sports activities have been identified in Gura Râului and Rășinari, and the weekly operation of a local producers' market is mentioned in Rășinari, where efforts to set up producers' associations are also noted.

In order to identify the perception of the respondents regarding the quality of life and services at the locality level, the differential semantic scale was used (Table 9). The answers were collected using a Likert scale with 4 steps (3, very satisfied; 0, not satisfied at all). The results obtained allow us to measure the degree of satisfaction of the respondents regarding the quality of life in the locality, safety, quality of education and health services, and quality of cultural and leisure services.

**Table 9.** Quality of life and services score.

|  | Score | Rank |
|---|---|---|
| Quality of life in the locality | 2.84 | 2 |
| Safety at local level | 2.97 | 1 |
| Quality of education services | 2.19 | 4 |
| Quality of health services | 1.63 | 5 |
| Cultural and leisure services | 2.81 | 3 |
| Average score | 2.49 |  |

Source: Own results based on social survey.

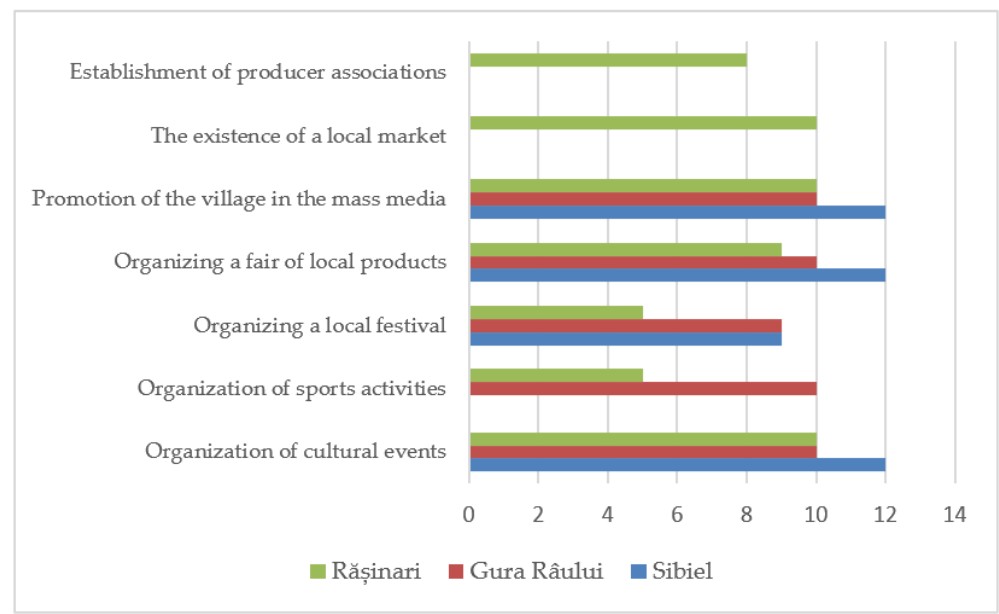

**Figure 6.** Measures taken by local authorities for the economic development of the locality. Source: own design based on field research.

The overall score of 2.49 for the perception of the quality of life and services in the locality shows that the managers of tourist accommodation facilities consider both the quality of life and services to be of a high standard. The highest value of the calculated score (2.97) was obtained for the safety of life at the locality level. Furthermore, in the Romanian countryside, there are some unwritten rules of behavior, because "everybody knows everybody". The lowest calculated score was obtained for the quality of medical services, an aspect that needs improvement in all three localities.

With the help of the differential semantic scale, the degree of satisfaction of the respondents regarding the social life in the locality was also identified (Table 10). The answers were collected using a Likert scale with 5 steps (4—very satisfied; 0—not at all).

**Table 10.** Satisfaction regarding some aspects of social life in the locality.

|  | Score | Rank |
| --- | --- | --- |
| The involvement degree of residents in community life | 3.13 | 1 |
| Number of community organizations | 2.69 | 4 |
| Turnout in the elections | 3.09 | 2 |
| Degree of involvement in local development projects | 3.06 | 3 |
| Average score | 2.99 | |

Source: Own results based on social survey.

In all three rural communities, there was social life, with local people feeling involved in the smooth running of development activities (score 3.13). However, there is a need for new community organizations to respond to citizens' needs (score 2.69).

4.2.5. Identifying Innovative Technology Solutions Used in Tourist Facilities and Food Supply for Tourists

In all the tourist facilities analyzed in the three localities, 100% have internet access, 65.60% use camera surveillance systems, 40.60% have installed solar panels for domestic hot water production, 37.5% have photovoltaic panels for electricity production, and 6.25% use smart farming solutions (drones, GPS, animal sensors).

The access of tourists to smart solutions at the level of the tourist structure shows that 100% of them benefit from fast internet, access to modern digital equipment (smartphones,

laptops, tablets) and possibilities to connect to e-commerce services (telemedicine, online booking, e-learning).

The main sources of food served to tourists are the farms of the producers in the locality or neighboring areas (96.87%), supermarkets (87.50%), shops of local producers (84.38%), their own production (78.13%), and local markets (62.50%).

4.2.6. Identifying the Perception of the General Appearance of the Locality and the Efforts Made to Preserve the Natural and Anthropic Tourist Heritage

On a scale of 1 to 5 (5—total agreement; 1—total disagreement) all respondents gave maximum scores to the following 8 aspects related to the tourist village: the locality has the image of a healthy and attractive village; a marketing strategy is implemented in the locality to promote the natural and anthropic tourist heritage; efforts are made in the locality to preserve and promote cultural traditions, and artistic events are organized, which are integrated in the tourist programs and attract tourists; the cultural heritage is well preserved and represents an attraction point for tourists; villagers of all ages dress up in folk costumes for different events; there are numerous films and TV spots broadcasted on mass media channels promoting the attractiveness and image of the village; there is local concern for creating a close link between the appearance of the locality, the cultural tradition, and the natural landscape; and there are marked hiking trails around the locality which are used by locals and tourists for outdoor activities and observation of local flora and fauna. The maximum score awarded leads us to state that local tourism entrepreneurs are satisfied with the image of the tourist village and appreciate the efforts made by the local public administration to promote it.

5. Discussion

With an area of 71,340 km$^2$, the mountain area in Romania represents 29.92% of the country's surface (238,391 km$^2$), and it is the living environment for 3.35 million inhabitants (24.92% of the country's population). In total, 37.8% of the administrative territorial structures in Romania are located in the mountain area, including 80 towns and 3560 villages, in which there are more than 850,000 traditional households [63]. The constraints faced by this area are related to relief, depopulation, an ageing population, poorer living conditions, unsatisfactory services or lack of them, limited use of agricultural land, and a decline in livestock. For all these reasons, the mountain area is considered a disadvantaged area, being an area of national, strategic, economic, social, and environmental interest that requires sustainable and inclusive development. A major direction of development is that linked to sustainable tourism in all its forms [64].

Rural tourism and agritourism in Romanian villages is more than 30 years old, dating back to 1990, when the tourism sector was coordinated by the National Tourism Organization. In 1972, the Institute for Tourism Research identified 118 rural localities with tourism potential, of which 14 were declared "villages of tourist interest". Immediately after 1990, the Mountain Area Directorate of the Romanian Ministry of Agriculture established the first criteria based on which mountain households wishing to facilitate tourism could be classified. Subsequently, the classification of guesthouses was performed according to the Order of the Ministry of Tourism no. 20/1995 and the H.G. 58/1997. Ordinance 63/1997 referred for the first time to agritourism guesthouses and the facilities granted to owners. In the present day, the classification of tourist facilities is performed according to the Order of the President of the National Authority for Tourism No 65/2013 [65,66], amended and completed by Order 510/2022.

In 1994, the National Association for Rural, Ecological and Cultural Tourism (ANTREC) was founded, and it has made a major contribution (by organizing events and courses for tourism entrepreneurs, editing publications, exchanging best practices, etc.) to the promotion of rural tourism [66]. A year later, ANTREC Romania joined the European Federation of Rural Tourism (RuralTour) [67] and thus started to promote Romanian villages abroad. Other organizations that have contributed over time to the development of tourism

(with its various forms) in Romanian rural areas are Operation Village Romaine, Romanian Federation for Mountain Development, Federation of Mountain Farmers—Dorna, Romanian Ecotourism Association, Adept Foundation, and "Mihai Eminescu Trust" Association [68,69].

All these organizations have created various programs for rural development through tourism. Over more than 30 years, the number of tourist structures in rural areas has increased from approx. 500 units in 2000 to over 2500 in 2017 [69]. Rural tourism in mountain villages has made a major contribution to the diversification of economic activities and is considered a "smart chance" for local communities [33].

Adamov et al. show that in 2021, the top three rural tourism/agritourism destinations in Romania are the Brașov area, the Maramureș area, and the Sibiu area. The number of tourist structures and the net capacity utilization index registered in the three areas were 387 and 31.4% (Brașov); 264 and 26.4% (Maramureș); and 146 and 35.6% (Sibiu), respectively [70]. In the same year, 9146 tourist structures with accommodation functions were operating in Romania, of which 3460 were agritourism guesthouses, representing 37.7% of the total [71], registering an increasing demand for rural holidays based on the sustainability and uniqueness of the area [69]. Davidescu et al. also identified the main development pillars of rural tourism in Romania, based on a composite tourism development index. According to this index, Suceava, Harghita, Brasov, Argeș, Mures, Sibiu, and Cluj counties are at the top of the ranking [72].

The concept of HS covers all aspects of life in the countryside, and its implementation differs from one area to another [23], depending on the involvement of various local actors (guesthouse owners, farmers, local authorities, tourism agencies, tourism service providers) [73] and the partnerships that are created between them. It includes the general appearance of the locality, the quality of life, food, environment and services, the existence of high-speed internet and the digital skills of local people, the use of renewable energy sources and innovative technologies, local governance, etc. The results obtained are in line with those presented by Rahoveanu et al., 2022, who show that the use of the internet is essential to halt rural decline, the adoption of different types of renewable energy leads to energy independence of communities, and the multidisciplinary SV concept with all its aspects facilitates the development of rural areas [17], adding value to ecologically, economically, and socially sustainable communities. In this context, rural tourism and agritourism represent a "smart" opportunity for the sustainability of the rural environment in mountain areas, being considered a "smart" tool at the disposal of local communities [74]. The results obtained show that the owners of guesthouses appreciate the effort of the local administration to organize different cultural events that increase the attractiveness of the area, an aspect also confirmed by Stanciu et al., 2014 [75].

The rural tourism in Sibiu County is mainly developed in the villages in the mountain area and allows the enhancement of all natural and man-made tourist resources of the area. According to data published by the Romanian Ministry of Entrepreneurship and Tourism, at the end of 2022, 779 tourist structures with accommodation functions were operating in Sibiu County, of which 263 were the county's town (about 33.8%), 113 in other urban localities (14.5%), and 403 (51.7%) in rural areas [59].

The rural localities in Sibiu County with the highest number of tourist accommodation structures are Cârțișoara (38), Gura Râului (33), Sibiel (31), and Rășinari (29). All these localities are in the mountain area, and the last three belong to the "Mărginimea Sibiului" area, whose uniqueness is due to the ancient shepherding activity of the inhabitants. The general appearance of the villages, traditions, customs, local gastronomy, and the way of life of the locals derive from this.

The large number of tourist structures in the three localities chosen for the case study demonstrates the entrepreneurial spirit of the locals who see in rural tourism and agritourism an opportunity for future development based on the capitalization on natural and anthropic resources [70], making Mărginimea Sibiului a unique rural tourism destination. Coroș et al. show that since 2010, the efforts of the owners of tourist structures in the

Mărginimea Sibiului area have been observed to improve the quality of services and the level of comfort offered, which has led to the improvement of the attractiveness of the area [76]. Small-scale tourist structures, managed by local families, meet the current needs of tourists to enjoy privacy, sanitary safety, and social distance [77].

In all three localities, many agricultural activities have been identified as future options for local economic development, and most guesthouse owners choose to obtain their food from local farms or producers' shops. In this way, the whole community supports agriculture and is concerned about producing healthy food and creating a food system based on local production and traditional production practices. All these aspects have also been highlighted by Slee in 2020 [9], which exemplifies the link between the SV and the European Green Deal. Guesthouse owners know that tourists prefer the gastronomy specific to the area, based on tasty, fresh, and healthy food [78].

In the three localities, efforts are being made to promote the natural and anthropogenic tourism heritage and attract tourists by organizing local festivals and fairs; organizing a weekly local farmer's market; making available to tourists the "The road market" mobile phone application for buying food products at the farm gate and the "Questo" application for identifying tourist destinations and routes; booking accommodation through different platforms (Booking, Travelminit; Trip Advisor; Turist info; Direct booking; "At boarding houses"; Tourist Romania; Sky trip; Tourism guide), web pages of the guesthouses. Many of the guesthouses have open accounts on the social networks Facebook and Instagram, which facilitates promotion and the possibility of booking.

The local communities in the three villages have managed to preserve and enhance their cultural heritage and have integrated technology into everyday life and for tourism purposes. Tourism thus creates opportunities to diversify the local economy and stimulate rural development.

The future development of the pastoral villages of Mărginimea Sibiului, including the three chosen in this case study, must be based on the revitalization of extensive sheep farming practices, the preservation of the pastoral economy, and the development of tourism [79,80] in close connection with the specificity of the area.

## 6. Conclusions

Smart tourist villages must be self-sustainable villages, with communities involved in traditional activities, specific to the local culture, because tourism is the crowning glory of the rest of the community's activities and is based on strengthening the community. Each tourist village must identify its own brand and the values that are the basis of it.

We believe that the biggest challenge for mountain tourist villages in the Sibiu Marginal area is the need to involve the whole community and especially the younger generation in preserving traditions and extensive agricultural practices.

To increase the interest, motivation, and involvement of the local population, it is necessary to organize informal meetings with the participation of representatives of interest groups, the so-called "informal village parliament". In such meetings, the needs of the community will be identified, and activities can be designed to address them.

Rural tourism in mountain villages contributes not only to preserving the cultural heritage and rural lifestyle but also to raising awareness among locals and tourists of the value of the natural and agricultural heritage. In all these communities, tourism must highlight the results obtained from extensive farming and livestock farming, with an emphasis on the nutritional and organoleptic quality of the products obtained. Registering as many mountain food products as possible in various quality schemes and promoting their authenticity and multiple benefits revitalizes the agricultural and traditional raw-material-processing sectors. Mountain farmers need support from the state and the business environment to create short food chains through which to market their products. We believe that the setting up of more local gastronomic points in these villages would highlight traditional culinary dishes with local ingredients, prepared and served by locals. It is

important that all income from agriculture, food processing, and tourism must stay in the community and contribute to improving the quality of life of farmers.

The authenticity of the tourism offer is particularly important because tourists choose villages for the simplicity of life. Authentic tourism focuses on local culture, a fair economy, nature, and biodiversity, giving visitors the opportunity to connect with the essence of a locality. Tourists interact with the character and hospitality of the whole community, which requires openness, curiosity, and respect, without offending the dignity of the locals and while accepting cultural differences. By interacting with the local community, tourists can discover something different, specific to the chosen destination, for the purpose of knowledge and understanding.

Authentic tourism supports local initiatives to preserve local heritage and identity and contributes to awareness of their value. The authenticity of an area is preserved if there is a dynamic link and homogeneity between accommodation conditions and facilities, the natural setting, and the holiday experience in the community [81].

We believe that the younger generation, which is more technology-oriented, could be more involved in tourism, especially in creating a greater variety of tourism packages, including cooking workshops; tastings; leisure activities; educational activities providing information about the agricultural production system, traditional food products, and local culture; visits to farms and/or craft workshops; excursions to the surrounding area; mountain hikes; and folklore evenings. The development of such activities should be based on local hospitality and culture, integrating technology. Diversification of activities has many advantages for the local community, especially in terms of job creation and reduced seasonality. At the same time, it would create a comprehensive tourist experience, leading to a favorable perception [82].

The main limitation of the study is the evaluation of some mountain tourist villages (known for the area's tradition of sheep farming) only from the perspective of rural tourism entrepreneurs, without considering the perspectives of field employees or tourists.

The results obtained cannot be generalized to all mountain villages. Future studies are needed to also address this topic from the perspective of other categories of stakeholders, representatives of the economic environment, local public administration employees, or tourists. Furthermore studies could be conducted in the future making comparisons between tourist villages in different mountain areas of the country or outside it. Another possible direction of investigation is related to the use of smart technologies in peasant households in the mountain area, especially those that lead to the easing of household work. Other possible future research directions are the use of smart technologies in rural households in the mountain area with tourism, especially those that lead to the easing of household work; accessing funds from different sources to implement technology in agricultural/zootechnical activity; and satisfaction regarding the quality of life in the mountain area for the various stakeholders.

**Author Contributions:** All authors contributed equally to the research and writing of this study. Conceptualization, M.S. and G.M.; methodology, M.S.; software, M.S.; validation, A.P. and I.A.B.; formal analysis, M.S.; investigation, G.M.; resources, M.S., G.M. and I.R.; data curation, M.S.; writing—original draft preparation, G.M., A.P., I.A.B., I.R., B.G.N. and M.S.; writing—review and editing, M.S. and I.R.; visualization, G.M., B.G.N. and M.S.; supervision, A.P., B.G.N. and M.S.; project administration, G.M.; funding acquisition, G.M. All authors have read and agreed to the published version of the manuscript.

**Funding:** Project financed by Lucian Blaga University of Sibiu through the research grant LBUS-IRG-2022-08.

**Informed Consent Statement:** Not applicable.

**Data Availability Statement:** Not applicable.

**Acknowledgments:** The authors thank the guesthouse owners in the three villages.

**Conflicts of Interest:** The authors declare no conflict of interest.

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
