# Peer review of "Can We Talk about Smart Tourist Villages in Mărginimea Sibiului, Romania?"

_sustainability, doi:10.3390/su15097475_

Round 1
Reviewer 1 Report
Thank you for submitting this manuscript for publication consideration into Sustainability-Tourism, Culture & Heritage. The topic of Smart Tourism as related to rural areas is an important one. Here are some considerations to help you further improve your manuscript.
1. Please explain and verify how and why these three areas were selected for your study.
2. Your research stages do not include transportation, which is a vital component for rural economic development. Yet, later in your paper you devote much attention to this area of analysis. It is recommended that your front end of the paper be more accurately developed to express the true areas of exploration/analyses.
3. Your entire analysis is based upon only the perspective expressed by 'managers'. This tends to present a bias, since management and visitors (tourists) often differ in their perceptions of the issues you have explored. In order for this paper to be meaningful, input from actual visitors (travelers/tourists) needs to be incorporated. In this manner, an "Importance-Performance" could provide more accuracy in what was reported by a small number of managers.
4. Although 'depopulation' and 'aging' were initially identified as being major concerns, why wasn't this addressed in further detail in your semantic differential scales items, or in personal 25-minute interviews with industry management? Similar issues exist throughout the world as younger generations are basically tech-oriented and do not want to spend their lives in an agricultural setting where little opportunity is available. The question then becomes one of how to identify manual labor sources to deliver those necessary services required by the traveling public in today's global environment. As an example, review the vast amount of literature related to wine tourism and how the rural aspects of these locations have been addressed. Many barriers/constraints can be learned from conducting a review of this body of literature/academic, scholarly research.
5. You stated that the questionnaires used both 4-point and 5-point scales. Yet, your tables only depict 5-point scale results. Please clarify.
6. In terms of the total number of lodging units within these rural communities, how many units were 'available' for use? Typically units may be undergoing repairs and are removed from useable inventory.
7. What was the occupancy rate for the different classifications of units during the past 12 months? This would assist the reader in better understanding lodging performance. Merely providing numbers of units is not enough information to determine rural community actual needs.
8. What were the total numbers of visitors to each of these three destinations during the year prior to this research? What percent of these were day visitors vs. overnight visitors?
9. Why was there no pre-test conducted on the 22-itwem survey? How were these items validated as being reliable without a pre-test?
10. Table 4 is in an unreadable format.
11. This was a basic environmental scanning project using selected variables. Were there any political/governmental issues not explored that should be conducted in future studies? As an example, how would the revenue be generated to further develop what you have identified as necessary?
12. The 'Conclusions' section is abbreviated. More discussion is needed and it should devote attention to the study's limits and recommendations for future research addressing this topic.
Author Response
Thank you for your constructive review of the article. Find our answers in the attached file.

Reviewer 2 Report
A very interesting article on rural tourism activities. After reading the paper, I got the feeling of "to good to be true". I enjoy reading about how the three chosen rural village are doing, how they are doing, and how well they are doing. However, in the back of my mind, I get the question mark, if they are doing that great, what is the point? These villages were tourist attractions since the 1960´s. They been enhancing what they have, preserving the cultural heritage they poses, and selling the concept of local resources and products. The only thing that I got after reading the article, is that the village is only changing/adapting new technology to continue doing what they doing (and that’s very fine). I was expecting to see two opposite examples. Say a good SV versus one that is not that. How another not a good SV village can learn from the experience of those three SV.
In the conclusions, the authors mention that one of the biggest challenge is "is the need to engage the whole community in the management of the tourist heritage (line 547-548). Personally, I think is sending the wrong message, if the whole community do that, the cultural patrimony become a tourist show business, the cultural patrimony succumbs to the tourism demands. What the research illustrated is that these three SV had been successful because they work and preserve the cultural patrimony (material and immaterial) as it is, keeping it authentic. What the community need to do, is continue to be farmers, agriculture, what the state/private industry need to do is create new markets for those products (and yes, tourism is one of them, but not the only one). The COVID-19 crisis just show us that if we put all the eggs in the tourism basket, for the next health crises, we may lose the basket.
Table 4, need to be fix, the units of measurements are difficult to read.
Author Response

(The authors gave the same response as above.)

Reviewer 3 Report
Dear Authors,
Thank you for the opportunity to review the article "Can we talk about Smart Tourist Villages in Mărginimea Si-2 biului, Romania?". The article is well-structured, frames the research questions well, and adds new information - through a case study - that can have scientific impact and contribute to the advancement of knowledge and research associated with tourism, particularly in rural or mountain contexts.
The overall evaluation is clearly positive and favorable for publication. However, the authors should consider the introduction of minor revisions that can help improve the reach and coherence of the text, specifically:
1- In the introduction, the demographic/social component related to the depopulation process that characterizes much of rural areas in Europe is mentioned. In the literature review, several ideas related to tourism and its territorial relationship are explored. It would be important to deepen this relationship, namely through a systematization of the advantages that tourism (in its multiple forms) can have for territories in decline, as well as the risks it incorporates.
2- In the presentation of results, it would be important to have a contextualization of tourism in Romania. How has it evolved in general, and particularly in the context of mountain villages?
3- The discussion/conclusion is somewhat vague and poorly substantiated. For example, what is authentic tourism? What is the authenticity of a territory? Is there a future (with tourism) for all territories in demographic decline? These and other questions should be further explored, in relation to the research questions presented earlier and with possibilities for future research/intervention.
In my opinion, these three developments will increase the quality of the text, which already presents great value and excellent organization and research
Author Response

(The authors gave the same response as above.)

Round 2
Reviewer 1 Report
Dear Authors,
Thank you for your revised document. Although some improvements can be seen in the paper, there are still areas needing improvement.
1. You need to address in your study limitations that the research only focused upon the viewpoints of conveniently selected industry lodging managers. No information was obtained from employees, or most importantly, visitors. Viewing research through the lens of just management tends to present a study bias that must be recognized in the 'limitations' section.
2. Please address why no information was presented on the semantic differential scales using 4-point measurements. Use of 4-point scales are criticized because they lack a mid-point. Please explain.
3. Occupancy rate vs. overnight use: The presented information validates that there is an extremely low occupancy rate reported in the three study areas. Typically, a break-even point of occupancy represents a rate of about 66%. The reported occupancy rates are more than 50% below this industry standard. This is why it is so important for this research to obtain the perspectives of visitors using accommodations.
4. Visitor numbers are very low. Since the research addresses Smart Tourism, discuss what is currently being used to communicate to potential visitors via Smart Tourism. Without asking visitors questions related to where their origin may be, explain how Smart Tourism can be used to help address the apparent need for more 'target marketing' efforts to increase visitor numbers and overnight guests.
5. The Conclusion section repeats information regarding the three selected study sites unnecessarily. Remove this from the Conclusions section.
Some minor edits are needed for English. Do not worry about this as he editorial staff will finalize the document if accepted.
Author Response
Our answers can be found in the attached file.
